# Soil Moisture Contribution to Winter Wheat Water Consumption from Different Soil Layers under Straw Returning

Lishu Wang [1,2], Xiaoxiang Zhou [1], Yumiao Cui [1], Ke Zhou [3], Changjun Zhu [1] and Qinghua Luan [4,*]

1   School of Water Conservancy and Hydroelectirc Power, Hebei University of Engineering, Handan 056038, China; fc8232@126.com (L.W.)
2   Hebei Engineering Technology Research Center for Effective Utilization of Water Resource, Hebei University of Engineering, Handan 056038, China
3   China Institute of Water Resources and Hydropower Research, Beijing 100038, China
4   Cooperative Innovation Center for Water Safety and Hydro Science, Hohai University, Nanjing 210098, China
*   Correspondence: carol97011202@163.com; Tel.: +86-159-327-57-109

**Abstract:** To study the contribution of moisture from different straw-treated and irrigated soil layers to the water consumption of winter wheat in dry farming, a 2-year straw treatment and regulated deficit irrigation experiment was implemented. The field experiment was carried out with 0% (S0), 1% (S1), and 2% (S2) straw returning amounts, and 75 mm (V3), 60 mm (V2), and 45 mm (V1) irrigation volumes. This experiment involved nine treatments, used to quantitatively analyze the ratio and variation of soil water use from different soil layers via the direct contrast method (DCM) and the multiple linear mixed model (MLMM). The results show the following: (1) The distribution of precipitation isotope compositions displayed a repeated trend of first decreasing and then increasing during the study period. Regression analysis showed that the local meteoric water line (LMWL): $\delta D = 6.37\delta^{18}O - 3.77$ ($R^2 = 0.832$). (2) With increasing soil depth, the $\delta^{18}O$ value decreased gradually, and the maximum $\delta^{18}O$ value of the soil water within each growth period was distributed at 10 cm. (3) Under the same irrigation amount, $\delta^{18}O$ increased with increasing straw return at 0–20 cm and decreased with increasing straw return at 20–80 cm. (4) The comparison results of the DCM and MLMM were consistent. During the jointing and flowering stages, 0–30 cm soil water was the main source of water for winter wheat. The contribution of soil water below 30 cm had a decreasing trend from the jointing stage to the flowering stage. The average contribution rates of the 0–30 cm soil layer during the jointing and flowering stages were 23.07% and 23.15%, respectively. These findings have important implications for studying the soil water cycle in the context of farming.

**Keywords:** winter wheat; soil moisture; stable isotope; IsoSource model; water sources

## 1. Introduction

The North China Plain is located in the temperate, subhumid, continental, monsoon climate zone, characterized by distinct seasonal variations in air temperature. Covering approximately 20% of the country's arable land area, it is an important grain production base in China [1,2]; however, its water resources only account for 3% of China's, making resource shortage a serious issue in this area [3]. Therefore, the sustainable development of traditional agriculture is facing a serious water crisis, directly threatening safe water reserves and food security in north China. The main problem of agricultural water use in north China is the low water use efficiency of farmland, along with the low coverage rate of water-saving irrigation, resulting in the inefficient loss of 30% to 50% of irrigation water, which has great potential for improvement [4,5]. It is of great importance to clarify the mechanism of the water migration and transformation of farmland ecosystems, as well as to improve the efficiency of agricultural irrigation by employing various measures to alleviate water scarcity related issues and realizing the sustainable development of agriculture [6,7].

Crop straw is the most abundant agricultural by-product in China and the world. China's annual crop straw resources account for about 30% of the world's total; however, large amounts of straw are burned and discarded, resulting in a waste of resources [8]. Straw returning can improve soil structure and fertility, promoting plant growth [9–11]. The mechanisms by which plants attain water are consistently among the topics researched in this research field. With the increasing climate change severity and water scarcity, in-depth research on the sources of plant root water absorption is crucial for understanding plant water absorption mechanisms, improving plant drought resistance, and optimizing water resource utilization. Studies have shown that plant roots can absorb soil water through a variety of ways, among which are the root hair, capillary force, and interception absorption methods. Traditional methods for studying plant water sources are mostly based on root digging, soil water distribution observation, and water potential measurements, but these methods are vulnerable to environmental factors. Additionally, using these methods, soil water consumption in plants is only calculated according to changes in the soil water storage and crop root distribution, making it difficult to reveal the details of water uptake by roots, such as their depth and their contribution to the overall water uptake of the plant [12–14].

The hydrogen and oxygen stable isotope method has received extensive attention in research into predicting water sources in forest, grassland, and farmland ecosystems [15,16]. In recent years, stable isotope technology has become an advanced and accurate method for determining plant water sources, due to its high sensitivity and accuracy [17–19]. A number of researchers have studied the contribution of plant water consumption sources through isotope labeling [20–23]. Most relevant studies have used the direct comparison method [24,25] and the equal source mixed model to determine plant water sources [26,27]. The linear mixing model can determine plant water source apportionment by solving a series of equations when the number of sources does not exceed the number of element isotopes [28,29]. When the number of water sources exceeds the number of element isotopes, the multi-water linear mixed model can be used to quantify water source allocation [30–32].

At present, the environmental isotope method is gradually being integrated into the study of the farmland water cycle, but the quantitative analysis of the movement law of the crop root zone with the coupling of the isotope method and numerical simulation is still not perfect, and relevant studies on the application of straw returning to the field has not been reported.

Therefore, allocating the North China Plain farmland irrigation area as the research area and typical crop winter wheat as the research object, this study calculated the water source used by winter wheat within each growth period by setting different straw returning and irrigation amounts. Furthermore, the contribution of soil moisture from different soil layers to the growth and water consumption of winter wheat was studied to elucidate the soil water use characteristics of winter wheat. Therefore, the purpose of this paper is to further promote research into water balance in dry farmland and provide technical guidance for agricultural production.

## 2. Materials and Methods

### 2.1. Overview of the Study Area

The experiment was conducted at the Mingguan Campus, Yongnian District, Handan City, Hebei Province (located at the longitude of 114°20′ E–114°52′ E and latitude of 36°35′ N–36°56′ N). The test site was located in the south of Hebei Province and north of Handan City (Figure 1). The terrain of the whole area is high in the west and low in the east. From west to east, it can be divided into regions with low mountains and hills, piedmont sloping plains, plains, depressions, and other geomorphological units, belonging to a warm, temperate, semi-humid, continental, monsoon climate. Between 2007 and 2017, the average annual precipitation in the study area was 503.6 mm; July–August was the main flood season, during which the rainfall was 285.1 mm, accounting for 58% of the annual rainfall. Winter (December–January) precipitation was 14.1 mm, accounting for only 3% of the

annual precipitation. From 2007 to 2017, the average annual evaporation was 1997.5 mm, with the peak being in June (average of 354.4 mm) and the trough in December–January (averaging 53.6 mm) (Figure 2).

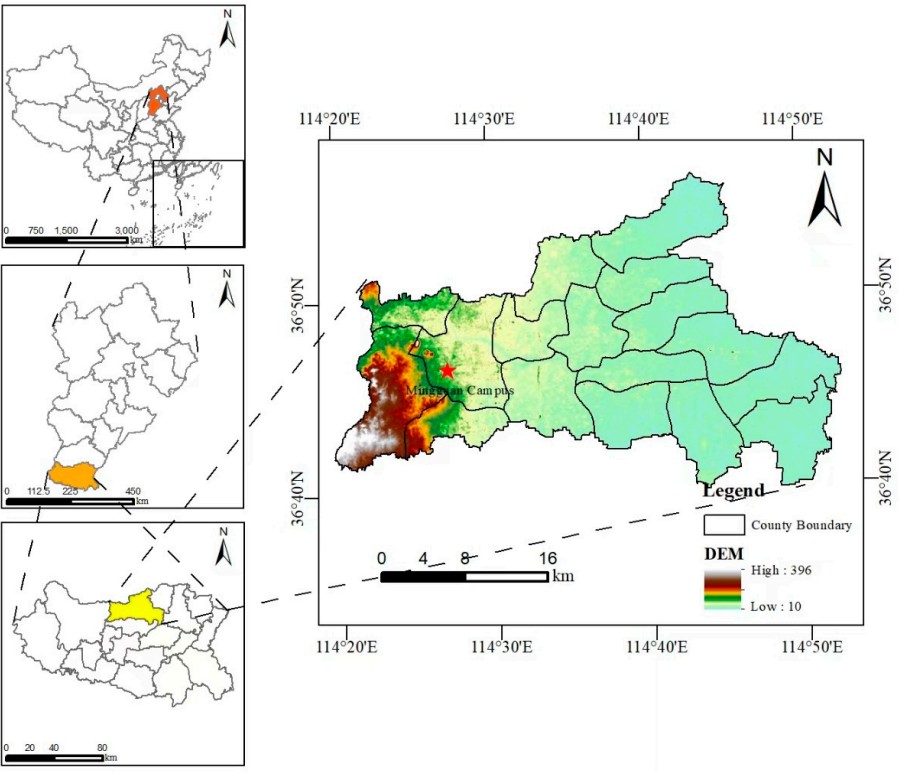

**Figure 1.** Geographical location of the experimental site (DEM: digital elevation model).

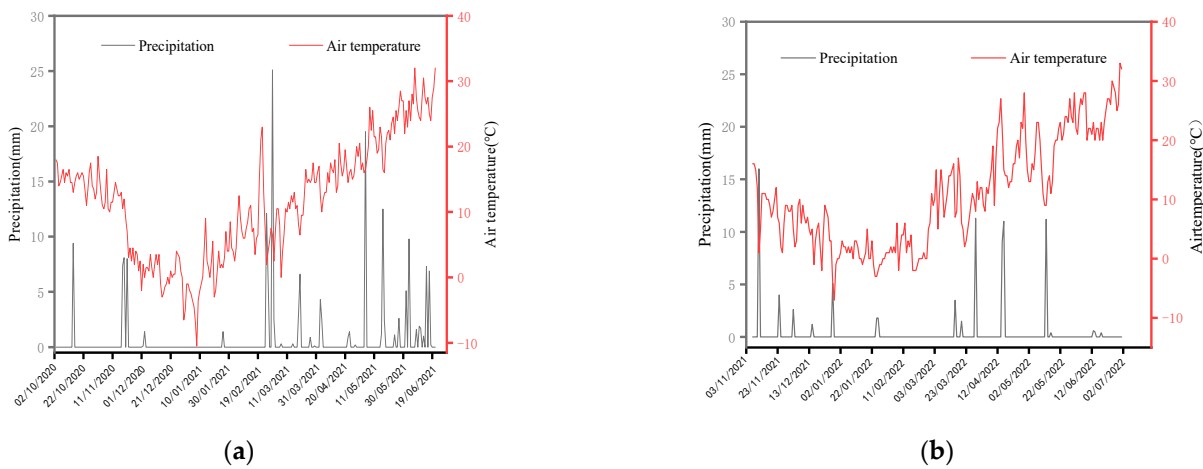

(**a**)                              (**b**)

**Figure 2.** Daily precipitation and temperature data for the experimental site from 2 October 2020 to 19 June 2021 (**a**) and 3 November 2021 to 2 July 2022 (**b**).

## 2.2. Experimental Design

The research area was chosen to be the farmland irrigation area of the North China Plain. According to the local farmland and soil conditions, different straw returning applications and irrigation water amounts were selected. Corn straw was crushed as the treatment for returning straw to the field. In all experiments, a unified lower limit (60% of field water retention) was used to control irrigation time, and the tillage treatment was the same as that of the local field. In the experiment, 0%, 1.0%, and 2.0% straw return methods (S0, S1, and S2) were set up. The unreturned straw was removed, and the replaced straw

was evenly covered on the soil surface by mechanical crushing at a ratio of 1% or 2%, then mixed with the soil. The irrigation amounts were 45 mm, 60 mm, and 75 mm (V1, V2, and V3). The area of each treatment was 120 m² (20 m × 6 m). There was a total of 9 treatments, and each treatment was carried out with 3 replicates. Field ridges were built around the area with a spacing of 1 m. The experimental design is shown in Table 1.

**Table 1.** Experimental design.

| Irrigation Volume | Straw Returning | | |
|---|---|---|---|
| | 0% (S0) | 1.00% (S1) | 2.00% (S2) |
| 75 mm (V3) | Area 3 (S0V3) | Area 2 (S1V3) | Area 1 (S2V3) |
| 60 mm (V2) | Area 6 (S0V2) | Area 5 (S1V2) | Area 4 (S2V2) |
| 45 mm (V1) | Area 9 (S0V1) | Area 8 (S1V1) | Area 7 (S2V1) |

All processing codes are in brackets.

### 2.3. Determination Items and Methods

#### 2.3.1. Soil Sample Collection

Soil samples of a 0–100 cm profile were collected using a soil drill from various layers (0–20 cm, 20–40 cm, 40–60 cm, 60–80 cm, and 80–100 cm) within a marked quadrate on 14 April (regreening stage), 26 April (jointing stage), 3 May (flowering stage), 15 May (filling stage), and 31 May (harvesting stage) of 2021 and 2022. A portion of each sample was sealed and stored in the laboratory, and the soil water samples were extracted using the Li-2000 low-temperature vacuum extraction system. The extracted water samples were sealed in glass water sample bottles, numbered, recorded, and refrigerated at 4 °C for testing. One part was placed in an aluminum box, and the soil moisture was determined using the drying method (105 °C, 12 h).

#### 2.3.2. Water Sample Collection from the Wheat Stem

During the wheat growth period, the sampling frequency of the crop stems was the same as that of the soil samples. Between 3 and 5 wheat plants were collected each time, and fresh stems close to the surface of the crops were collected. The epidermis in direct contact with air was removed. The low-temperature vacuum distillation and extraction device extracted the water samples from the stem and put them into a large test tube, cooled by liquid nitrogen (10 min) and vacuumed with a vacuum value below 9.9 Pa. The test tube was heated, and the condensing tube was put into a liquid nitrogen cup and extracted for 1.5 h.

#### 2.3.3. Precipitation Sample Collection

An automatic rain gauge was used to collect the precipitation samples and record precipitation.

#### 2.3.4. Determination of Hydrogen and Oxygen Stable Isotopes

Wheat water sample $\delta D$ value determination: The $\delta D$ values of the wheat water samples were determined using an LGR LDT-100 liquid water isotope analyzer with an accuracy of $\pm 0.5\%$. The $\delta^{18}O$ values of soil, wheat, and precipitation water samples were determined employing the Finnigan MAT253, TC/EA method, with an analytical accuracy of $\pm 0.3\%$. In the formula $\delta D$ (or $\delta^{18}O$) = [(RSA − RST)/Rst] × 1000‰, RSA and RST represent the ratio of the stable hydrogen isotope D/H or stable oxygen isotope $^{18}O/^{16}O$ in the sample and the standard, respectively. The smaller the $\delta$ value is, the more depleted the heavy isotope is, while the larger the $\delta$ value is, the more enriched the heavy isotope is. The standard deviation is represented in the thousandths [25].

### 2.4. Data Processing and Analysis

The contribution of different levels of soil moisture to crop root water absorption was studied using direct comparison and the IsoSource multi-source linear mixed model (V1.3.1). The transformation process of irrigation water and soil water in different soil layers was analyzed. The water transport processes in the soil and crops under different treatments were also revealed [33,34]. The direct comparison method was based on the principle that different water sources have different isotopic ratios. The contribution rates of different water sources to winter wheat water consumption were calculated using a multi-source linear mixed model (IsoSource V1.3.1). The parameters of the source increment and mass balance tolerance in the IsoSource model were set at 2% and 1%, respectively. The figures in this work were graphed via Origin 8.1 (graphing and data analysis software, Northampton, MA, USA). Statistical analyses were conducted using SPSS 22.0 (SPSS, IBM, Chicago, IL, USA). SPSS 22.0 was used for analysis of variance (ANOVA) using the least significant difference (LSD) method ($p < 0.05$).

## 3. Results

### 3.1. Characteristics of the Hydrogen and Oxygen Stable Isotope Composition in Precipitation

Figures 3 and 4 show the composition and distribution of δD and $δ^{18}O$ in precipitation and the distribution of precipitation and $δ^{18}O$ and δD values during the study period, respectively. Atmospheric precipitation is an important component of the regional water cycle, and as the driving force of interaction and transformation it has a crucial impact on the water cycle process. The composition of and variation in $δ^{18}O$ and δD in precipitation are particularly important for exploring the composition of and variation in $δ^{18}O$ and δD within regional water bodies.

The δD and $δ^{18}O$ ranged from −90.17‰ to −20.02‰, and −12.52‰ to −2.51‰, respectively, with averages of −53.40‰ ± 21.45‰ and −7.61‰ ± 2.95‰. Regression analysis of the hydrogen and oxygen isotopes of rainfall was carried out to obtain the local atmospheric drawdown line. The local meteoric water line (LMWL): $δD = 6.37δ^{18}O − 3.77$ ($R^2 = 0.832$). Compared with the global meteoric water line (GMWL), the slope and intercept of the LMWL formula in this study area were both smaller, indicating that the evaporation fractionation phenomenon of the raindrops in this region was more intense when they broke away from the clouds and fell to the surface.

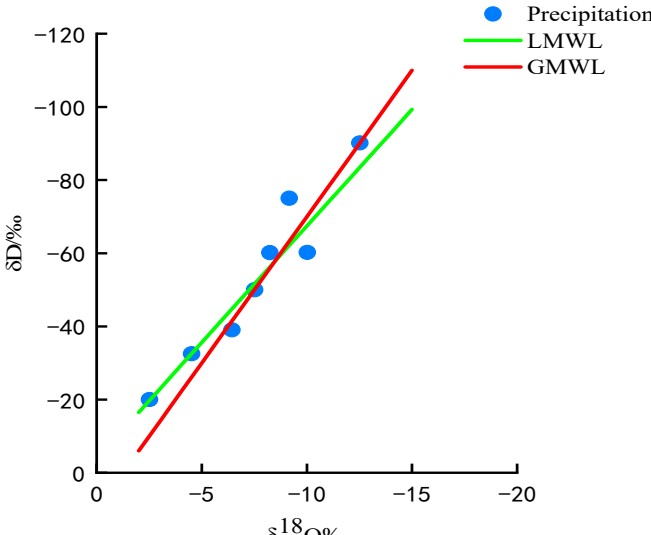

**Figure 3.** Relationships between the local meteoric water line (LMWL), global meteoric water line (GMWL), and precipitation $δ^{18}O$ and δD values from 2020 to 2021.

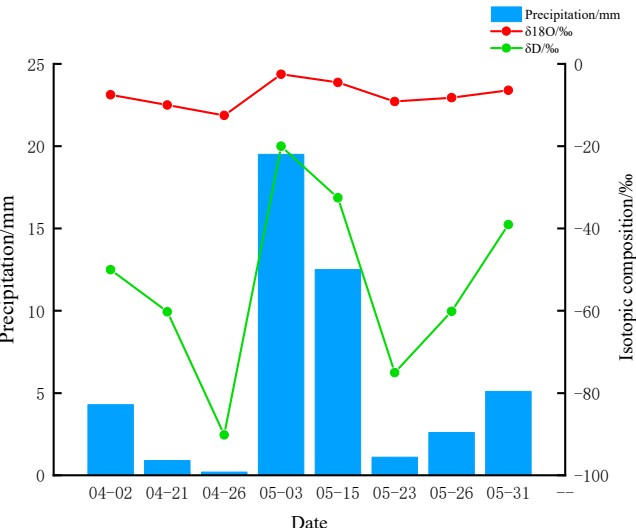

**Figure 4.** Precipitation and $\delta^{18}$O and $\delta$D values of precipitation during the study period between 2020 and 2021.

### 3.2. Effects of Different Straw Returning and Irrigation Amounts on Soil Water Content

As seen in Figure 5, the soil water profile of 0–1 m in the wheat sample plot showed inconsistent changes from the green stage to the harvest stage. Under different treatments, the water content of wheat during the jointing and flowering stages was relatively low, at 6.18% and 8.74%, respectively. The amount of straw returned to the field and the amount of irrigation affected the distribution of the soil water content. Under the same straw return amount, the soil water content decreased gradually with the decrease in irrigation amount. The soil moisture contents during the jointing and flowering stages were 3.61% and 1.05% lower, respectively, than the wilting coefficient, indicating an increase in water consumption during the jointing and flowering stages of winter wheat; that is, the water consumption of winter wheat increased during these growth stages. With an increased irrigation amount, the soil water content and storage also increased under the same straw returning amount. For example, the average water contents of areas 1 (S2V3), 4 (S1V3), and 7 (S0V3) at the jointing stage were 10.7%, 8.6%, and 0.3%, respectively. At the same time, under the same irrigation amount, with the increase in straw returned to the field, the average water content also increased. For example, the average water contents of areas 1 (S2V3), 2 (S1V3), and 3 (S0V3) at the jointing stage were 10.7%, 6.8%, and 6.5%, respectively. However, as the amount of straw incorporation into the field decreased, the average soil water content of area 3 (S0V3) tended to increase under the same irrigation volume.

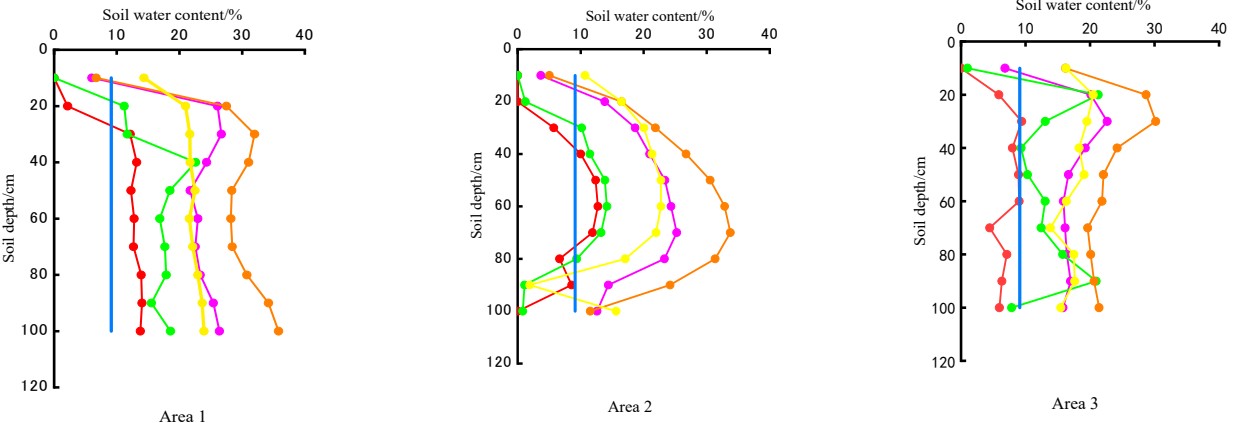

**Figure 5.** *Cont.*

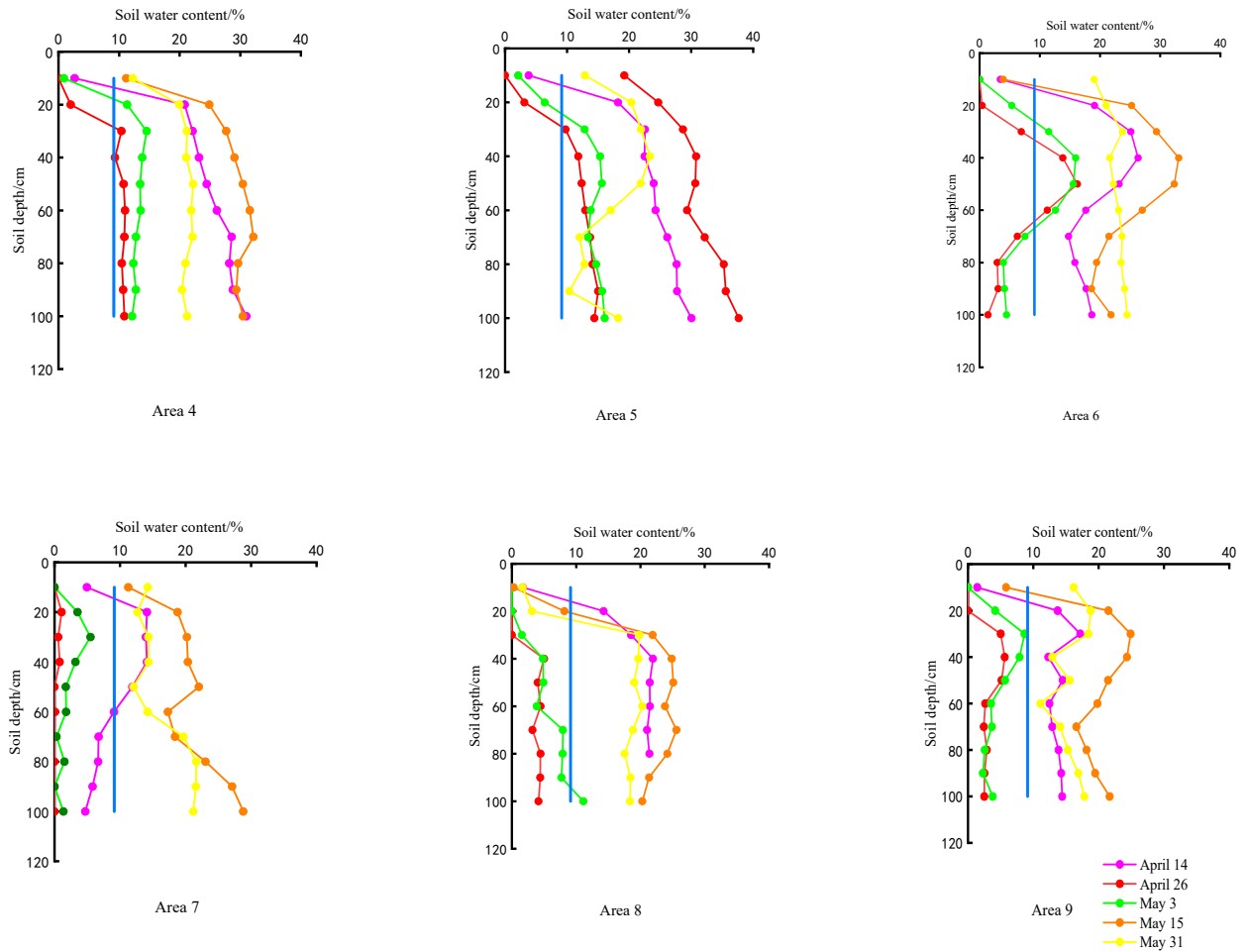

**Figure 5.** Dynamic changes in the soil fill water profile in winter wheat plots.

### 3.3. Effects of Different Straw Returning and Irrigation Amounts on $\delta^{18}O$

Figure 6 indicates that the straw returning amount has an effect on soil water $\delta^{18}O$ at different growth stages and with the same irrigation amount (because $\delta D$ and $\delta^{18}O$ are in good agreement with soil water isotopes). At different growth stages of wheat, $\delta^{18}O$ values decreased with the increase in soil depth. The effects of straw returning and irrigation amounts on $\delta^{18}O$ were different at the developmental phases of winter wheat. The maximum $\delta^{18}O$ value of soil water in each growth stage was distributed at 10 cm. Under the same irrigation amounts, from 0 to 20 cm of straw returning, there was a gradual rise in $\delta^{18}O$ values, whereas from 20 to 80 cm of straw returning, there was a gradual decline. Under the same straw returning amounts, $\delta^{18}O$ increased with the increase in irrigation(Table 2).

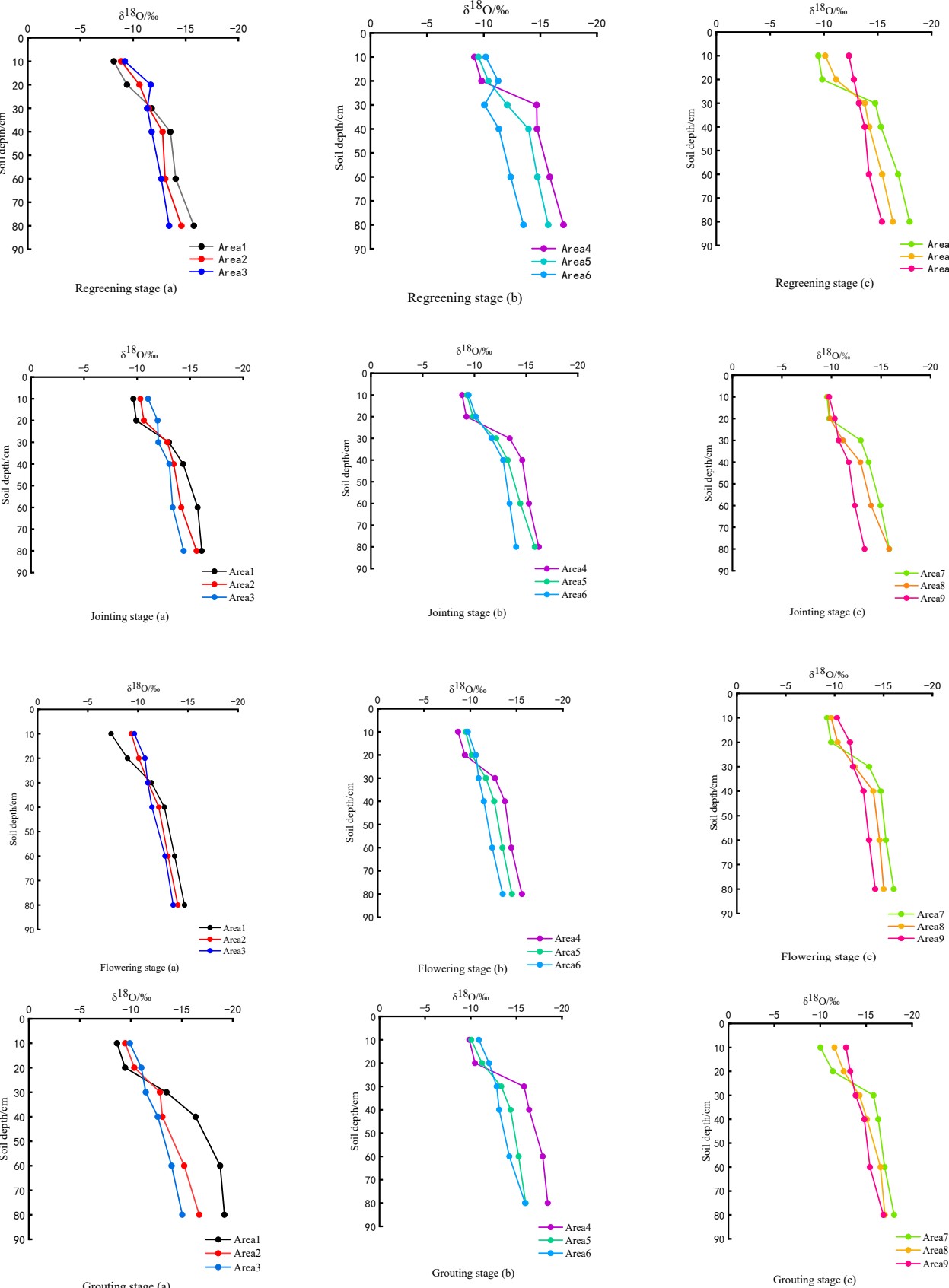

**Figure 6.** Changes in $\delta^{18}$O with soil depth under different periods and treatments.

**Table 2.** Effects of straw returning and irrigation amounts on soil water $\delta^{18}O$.

| Soil Depth/cm | Area 1/‰ | Area 2/‰ | Area 3/‰ | Area 4/‰ | Area 7/‰ |
|---|---|---|---|---|---|
| 10 | −8.17a [a] | −8.87b | −9.22c | −9.18 [b] | −9.48 [c] |
| 20 | −9.42a [a] | −10.61b | −11.68c | −9.81 [b] | −9.85 [b] |
| 30 | −11.78c [a] | −11.51b | −11.34a | −14.67 [b] | −14.77 [c] |
| 40 | −13.54c [a] | −12.80b | −11.77a | −14.71 [b] | −15.30 [c] |
| 60 | −14.05c [a] | −13.05b | −12.68a | −15.84 [b] | −16.90 [c] |
| 80 | −15.77c [a] | −14.59b | −13.43a | −17.05 [b] | −17.97 [c] |

Note: The letters in parentheses represent variance comparisons between area 1 (S2V3), area 4 (S2V2), and area 7 (S2V1), while the letters outside parentheses represent variance analysis comparisons between area 1 (S2V3), area 2 (S1V3), and area 3 (S0V3). Different lowercase letters indicate significant differences between treatments ($p < 0.05$).

Table 3 shows the effects of straw returning and irrigation amounts on the $\delta^{18}O$ of the stem water and water samples of winter wheat. As seen in Table 3, different straw returning and irrigation amounts resulted in the regularity of the $\delta^{18}O$ values of the water samples from each treated stem. From 14 April to 26 April, the stem water $\delta^{18}O$ of each treatment gradually increased. From 3 May to 15 May, the stem water $\delta^{18}O$ of each treatment gradually decreased. From 26 April to 3 May, the stem water $\delta^{18}O$ of areas 1 (S2V3), 2 (S1V3), 3 (S0V3), 4 (S2V2), and 5 (S1V2) gradually increased. The stem water $\delta^{18}O$ of areas 6 (S0V2), 7 (S2V1), 8 (S1V1), and 9 (S0V1) decreased gradually. During this growth period, the stem water $\delta^{18}O$ of each treatment had no significant difference. On the other hand, under the same irrigation amount, stem water $\delta^{18}O$ increased with the increase in the straw returning amount during the 14 April, 26 April, 3 May, and 15 May growth periods. With a constant straw returning amount, there was an upward trend observed in the $\delta^{18}O$ of stem water as the irrigation amount increased. This is mainly because with a greater amount of straw returned to the field, the effect of slowing down water became more obvious.

**Table 3.** Effects of straw returning and irrigation amounts on stem water $\delta^{18}O$.

| Period / Handle | 14 April | 26 April | 3 May | 15 May |
|---|---|---|---|---|
| Area 1 | −10.09 | −9.74 | −8.12 | −8.85 |
| Area 2 | −12.46 | −10.79 | −9.91 | −10.70 |
| Area 3 | −13.90 | −12.33 | −10.41 | −12.79 |
| Area 4 | −12.70 | −11.94 | −10.87 | −11.09 |
| Area 5 | −13.64 | −12.82 | −11.21 | −12.44 |
| Area 6 | −14.02 | −11.01 | −11.41 | −13.40 |
| Area 7 | −10.13 | −9.61 | −10.43 | −12.06 |
| Area 8 | −12.98 | −12.02 | −12.88 | −13.69 |
| Area 9 | −15.67 | −12.06 | −12.81 | −13.72 |

### 3.4. Contribution of Soil Moisture from Different Soil Layers to the Water Consumption of Winter Wheat

3.4.1. Direct Comparison Method

Precipitation can be absorbed and utilized by winter wheat only once converted into soil water. The direct comparison method is based on the principle that different water sources have different isotopic ratios. By comparing the isotopic compositions of soil water and plant water at different soil depths, as long as the stable isotopic compositions of the plant water and a layer of soil water are roughly at the same location or intersected, the plant prefers to use that layer of soil water. The soil water and plant water have similar isotopic values, indicating a larger proportion of soil water used from the soil layer [35,36]. Based on the isotopic characteristics of soil water and wheat stem water, the stable hydrogen and oxygen isotope profiles of soil water and their relationship with the

isotopic composition of wheat water were drawn. Due to the linear relationship between $\delta D$ and $\delta^{18}O$, the two results were similar. Figures 5 and 6 only show the $\delta^{18}O$ profiles. According to the direct comparison method, the main water source of wheat is the soil layer near the intersection of the soil water and wheat stem water isotope composition line. According to Figures 5 and 6, the possible main water source levels of winter wheat in the study area on April 26 and May 3 could be preliminarily obtained. On the other hand, the $\delta^{18}O$ intersections of the soil water and wheat stem water were different with the straw returning and irrigation amounts; that is, with the increase in straw returning amount, the $\delta^{18}O$ of stem water was larger and the intersections of the soil layer were shallower.

The $\delta^{18}O$ profiles of soil water shown in Figures 7 and 8 were analyzed. Under different treatments, the soil water within the soil layers of 10, 20, 30, 40, and 60 cm and their adjacent soil layers were preferably utilized in winter wheat at the jointing stage. In contrast, winter wheat at the flowering stage preferentially utilized soil water in the 20, 30, and 40 cm soil layers and their adjacent soil layers. During the period spanning from the jointing stage to the flowering stage, it can be inferred that when there is no dry soil layer, the primary water stratification source for winter wheat is mainly attributed to the shallow soil. There are two reasons for this: first, with the increase in rainfall and straw delaying water infiltration, the relative increase in shallow soil water content and water availability increased. Second, the wheat root system continues to develop further along with the growth period. However, the stem water of plants is a mixture of soil water from different soil layers and the direct comparison method assumes that plant roots preferentially use a specific layer of the soil water, allowing us to judge the approximate depth of soil water utilization by wheat, but not to quantitatively judge the contribution rate of the soil water from different soil layers to the wheat water consumption.

3.4.2. Multiple Linear Mixed Model

The multivariate linear model is a common method for determining the soil water source. It builds mathematical models by measuring multiple indicator variables in soil and plants to infer the source of water uptake by the plants. According to the distribution characteristics of the soil water isotope profile and the results of the direct comparison method, the 0–80 cm soil layer was divided into six groups: 0–10, 10–20, 20–30, 30–40, 40–60, and 60–80 cm. Then, according to the soil moisture content of each layer, the isotopic composition of each group of soil water was calculated by the weighted average. Finally, the water isotopic compositions of winter wheat and soil at the jointing and flowering stages were input into IsoSource software for calculation.

According to the results (Figure 9), regardless of whether at the jointing or flowering stage, the water consumption of winter wheat was primarily influenced by soil water in the 0–10 cm, 10–20 cm, and 20–30 cm soil layers, as indicated in Table 4. However, in plot 9 (S0V1), the contribution rate of the soil water at the 0–30 cm depth range was relatively low, mainly due to the small amount of straw returning and irrigation. Water was not efficiently stored in shallow soils.

Compared with jointing stage, the contribution rate of soil water to the winter wheat water consumption from the 0–10 cm and 30–40 cm soil layers at the flowering stage increased from 36.5% to 37.4% and 10.8% to 11.6%, respectively, while the contribution rate of soil water from other soil layers increased slightly. The average contribution rates of the 0–30 cm soil layer during the jointing and flowering stages were 23.07% and 23.15%, respectively. The outcomes of the multivariate linear mixing model were affected by factors such as the number of water sources, the isotopic composition of the water sources, and the mixing conditions.

The results of the direct comparison method and the multivariate mixed model are relatively consistent: the soil water within the 0–30 cm soil layer was the main source of water for winter wheat during the jointing and flowering stages, and the contribution of soil water from the soil layer below 30 cm from the jointing to flowering stages to winter wheat water consumption tended to decrease. The reason for this difference lies in the

high humidity and slow evaporation during the research period. Straw can store water to a certain extent, and shallow soil moisture undergoes significant isotope fractionation under evaporation. Reflected in the correlation between soil water isotope profiles and wheat water isotope composition, the two intersect, resulting in a misjudgment by the direct comparison method.

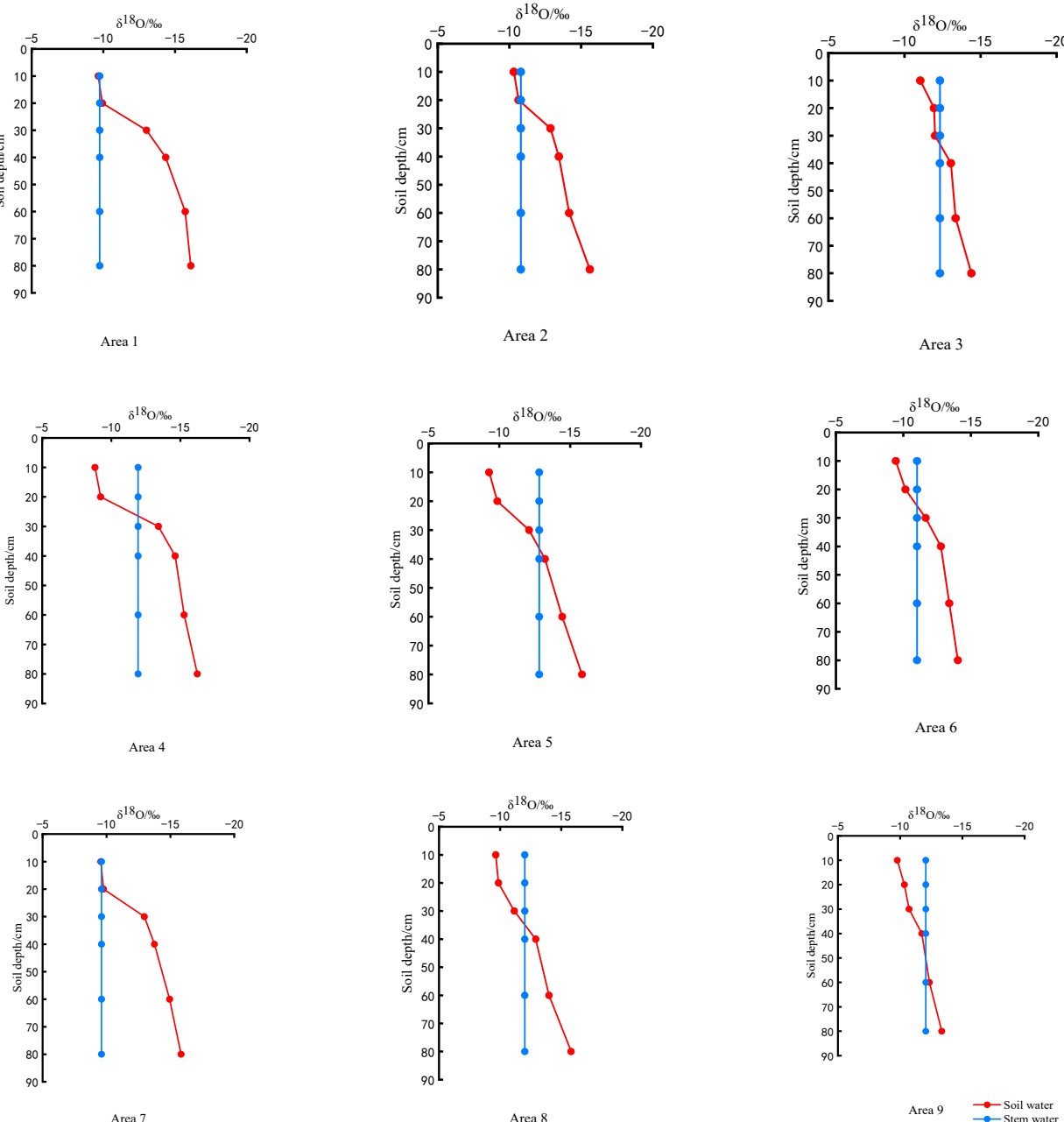

**Figure 7.** Soil fill water isotopic composition profile and wheat water isotopic composition at the jointing stage.

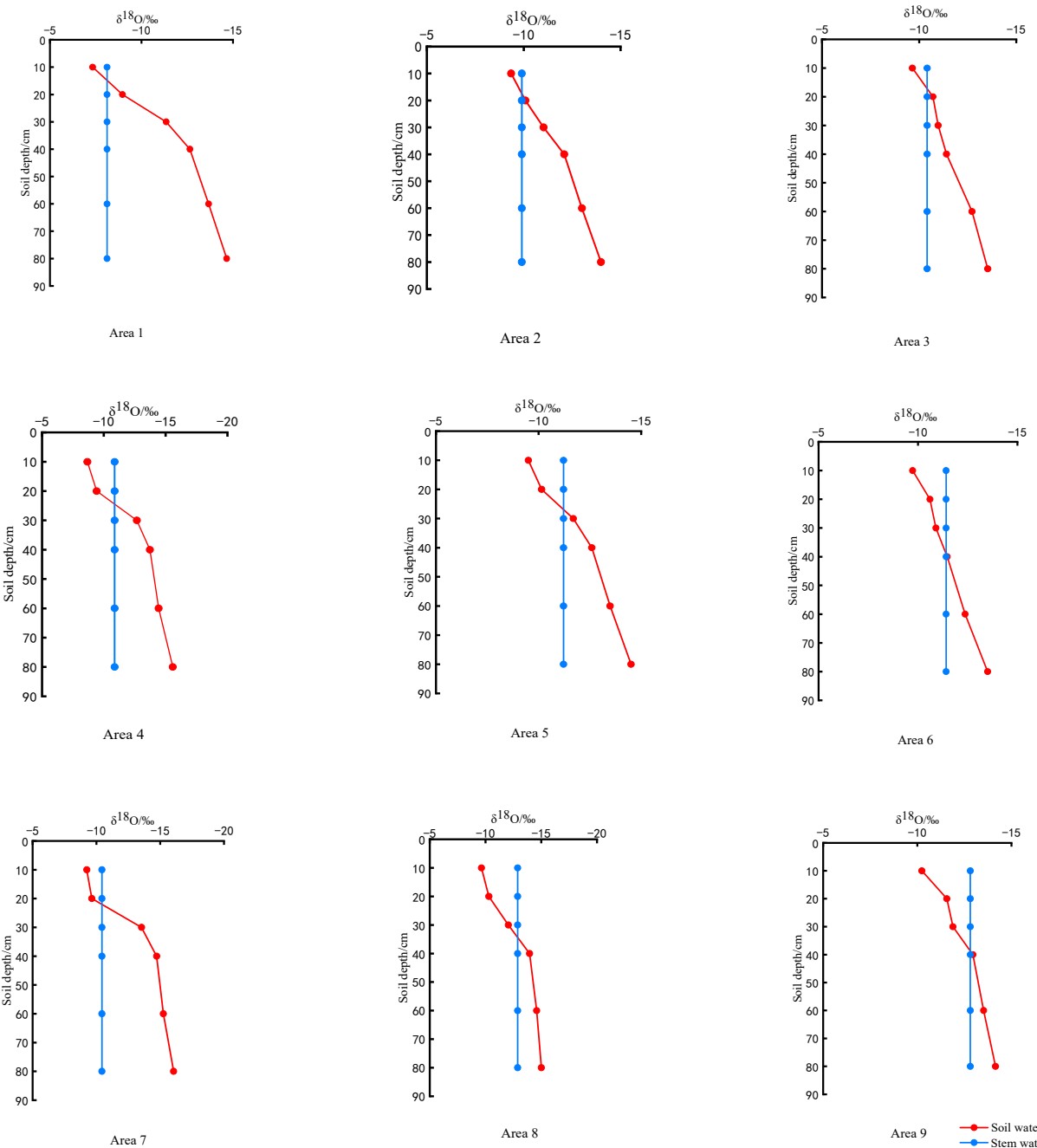

**Figure 8.** Soil water isotopic composition profile and wheat water isotopic composition at the flowering stage.

**Table 4.** The contribution of water sources from different soil layers to the water consumption of winter wheat under different treatments at the jointing stage (%).

| Water Source | Area 1 | Area 2 | Area 3 | Area 4 | Area 5 | Area 6 | Area 7 | Area 8 | Area 9 |
|---|---|---|---|---|---|---|---|---|---|
| 0~10 | 82 | 56.6 | 20.3 | 24.4 | 11.7 | 30.3 | 79.7 | 16.8 | 7.1 |
| 10~20 | 16.8 | 32.2 | 21.9 | 24.5 | 12.7 | 27.6 | 19.7 | 17.2 | 8.5 |
| 20~30 | 0.7 | 4.1 | 21.7 | 15.8 | 18.4 | 15.6 | 7 | 19.6 | 9.9 |

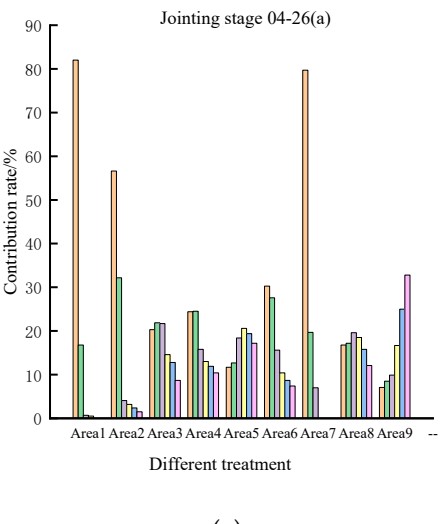

(**a**)

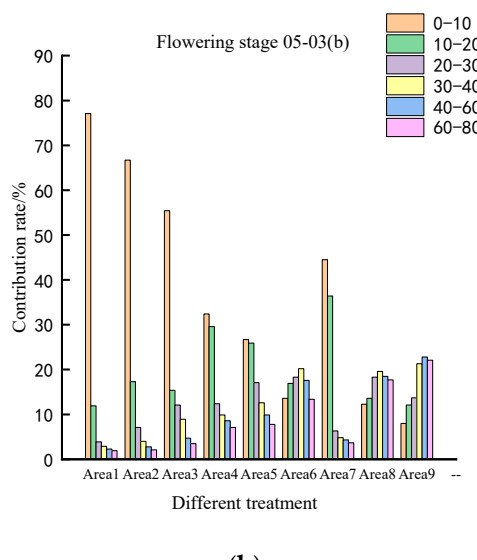

(**b**)

**Figure 9.** Contribution of soil water from different soil layers to the water consumption of winter wheat at the jointing (**a**) and flowering stages (**b**).

## 4. Discussion

As established, if soil water is sufficient, plants will grow fast, the population will be large, and the water consumption for growth and soil water consumption will increase accordingly. On the contrary, if soil water is insufficient, plant growth will be inhibited, which is not conducive to the increase in population, and the water consumption for growth and soil water consumption will also reduce accordingly [37]. Many studies have shown that straw returning can reduce soil ineffective evaporation and surface runoff, and it can improve the utilization rate of precipitation resources and soil water holding capacity, thereby increasing wheat grain yield and improving water use efficiency [8,9,37,38]. With the development of winter wheat, the root system is increasingly distributed to the deep soil. By the middle and late growth stages of winter wheat, the surface soil moisture content will be lower and lower, and water deficit will occur earlier [39–42].

Compared with the water sources (precipitation, soil water, irrigation water, and groundwater) available for winter wheat in shallow groundwater-buried areas and irrigation areas, due to the deep groundwater level in the study area, only precipitation and soil water could be used by winter wheat. At the same time, precipitation could be absorbed and used by winter wheat only once converted into soil water. Most studies suggest that returning straw to the field has a better water retention effect, and that the soil moisture content of wheat during the jointing and flowering stages is relatively low, at 6.18% and 8.74%, respectively. This may be because these developmental stages require large water amounts, but insufficient water supply leads to a decrease in soil moisture. The distribution of the soil water content was affected by the straw returning and irrigation amounts. The study revealed that when the soil moisture content was lower than the wilting coefficient boosting the irrigation quantity led to an elevation in the soil moisture content. With the same irrigation amount, increasing the amount of straw returned to the field can increase the average soil moisture content, which may be because it provides nutrients such as organic carbon, nitrogen, phosphorus, and potassium to the soil, improving the water and fertilizer retention capacity of soil, and thus the soil water use efficiency. However, as the amount of straw returned to the field decreases, the moisture content of some soils increases. It is speculated that the reasons are as follows: first, returning straw to the field slows down water migration, and the increase in straw returned to the field makes the phenomenon of water transportation more obvious. With the increase in irrigation amount, the water content under different soil depths increased. The maximum precipitation was only 19.5 mm, and it was difficult to penetrate to the depth of 20 cm [2]. The soil water con-

tent of the soil layer was lower than or equal to the wilting coefficient, so it was concluded that the soil water in this layer could no longer be used by winter wheat; winter wheat could only use precipitation. Second, the soil moisture content was lower than that of the capillary fracture (15%) and the soil evaporation was weak [38,43].

The stable isotope method provides a way to reveal the water absorption characteristics of plant roots due to their unique tracing ability [44]. The contribution of soil water to plant water consumption has been well-studied via isotope labeling methods. Yue used stable hydrogen and oxygen isotopes to analyze the isotopic composition of root active layer xylem water and soil water during the peak growth period of *Camellia oleifera* fruit, which is crucial for the scientific irrigation and ecological management of Camellia oleifera forests [27]. This study revealed the contribution of soil moisture from different soil layers to the growth and water consumption of winter wheat by setting different straw return and irrigation rates. Research has shown that the amount of straw returned to the field has a certain impact on the distribution of $\delta^{18}O$ in different water sources. In this study, under the same irrigation amount, the maximum $\delta^{18}O$ value of the soil water was distributed at 10 cm, and the effects of straw returning and irrigation amounts on soil water $\delta^{18}O$ value were different to some extent. The stem water $\delta^{18}O$ of each treatment had no significant difference. This is mainly due to the increased water requirements of growing winter wheat between the greenish and filling stages [35]. In the 0–20 cm soil depth range, the $\delta^{18}O$ value of the soil water increased with the increase in straw returning amount. This may be because a higher straw return increased the soil organic matter content and water retention capacity, so that more soil water came from precipitation, resulting in an increase in the soil water $\delta^{18}O$ value. However, the $\delta^{18}O$ value of soil water decreased gradually with the increase in straw returning amount at a 20–80 cm soil depth. This may be because the appropriate amount of straw returned to the field helps increase the vertical rise rate of soil water and reduces the phenomenon of water backflow in the deeper soil. In addition, increasing the irrigation amount under a consistent straw returning amount led to an elevation in the $\delta^{18}O$ value of soil water. This may be because the large amount of irrigation increases the input of soil water, bringing the $\delta^{18}O$ value of soil water closer to that of the irrigation water source. Based on the mass conservation of stable isotopes, $\delta^{18}O$ changes following a certain law. Regarding the jointing stage and irrigation amount, as the amount of straw returned to the field increased, the mean value of $\delta^{18}O$ of the soil water was low, but the mean value of the $\delta^{18}O$ of the stem water was high. For example, the mean values of the $\delta^{18}O$ of soil water in areas 7 (S2V1), 8 (S1V1), and 9 (S0V1) were 12.8 ‰, 12.2 ‰, and 11.4 ‰, respectively. The $\delta^{18}O$ stem water delta values for these areas were 9.6 ‰, 12.0 ‰, and 12.1 ‰, respectively. With the same straw returning amount, the $\delta^{18}O$ of soil water decreased with the increase in irrigation amount, while the $\delta^{18}O$ of stem water fluctuated. For example, the mean values of $\delta^{18}O$ of the soil water in areas 1 (S2V3), 4 (S2V2), and 7 (S2V1) at the jointing stage were −13.1‰, −13‰, and −12.8‰, respectively. On the other hand, the $\delta^{18}O$ values of the stem water in the same areas were −9.7‰, −11.9‰, and −9.6‰, respectively. The main reason for this was that straw stored more soil water that did not easily evaporate. During the elongation stage of winter wheat, the water consumption increased, so it was necessary to absorb water from soil to meet the needs of growth.

This study used the direct comparison method and a multiple linear mixed model to analyze the contribution of different straw return and irrigation amounts to winter wheat water consumption. The direct contrast method can qualitatively determine the approximate depth range of soil water used by wheat; using the IsoSource model, we can accurately quantify the proportion of water consumed by wheat derived from different soil layers [36,45,46]. In this study, the results of the direct comparison method and the multivariate linear mixed model were roughly the same. During the jointing and flowering stages, the major water supply for winter wheat came from the soil water present within the 0–30 cm soil layer, indicating that the plants mainly obtained water from the moisture at this soil layer depth and growth stage. Moreover, the contribution of soil water to the

water consumption of winter wheat from the soil layer below 30 cm at the jointing to the flowering stage showed a decreasing trend. This means that as the winter wheat growth progressed, the ability of deeper soil moisture to provide water to the plant gradually diminished. As precipitation was less available and straw was returned to the field, more water could not infiltrate the deep soil, which reduced the utilization intensity of water from deep soil by winter wheat, thus reducing the contribution rate of water from deep soil to the water consumption of winter wheat. Through field experiments and stable hydrogen and oxygen isotope techniques involving winter wheat in the Fengqiu area, Zhang found that the main water source for winter wheat during the tillering, turning green, and jointing stages is surface soil water, and the utilization of soil water from each layer decreases with increasing depth [47]. This is similar to the results obtained in this study.

There is a close relationship between the soil moisture content, soil water $\delta^{18}O$, and stem water $\delta^{18}O$ in winter wheat. By analyzing the soil moisture content, the irrigation volume can be adjusted in a timely manner to provide an adequate water supply. Analyzing soil water $\delta^{18}O$ and stem water $\delta^{18}O$ within different soil layers allows us to evaluate the source of soil water and the utilization efficiency of plants from different soil layers. This helps to carry out irrigation and soil water management scientifically and reasonably, thereby improving crop production and water resource utilization efficiency.

## 5. Conclusions

(1) During the study period, the isotope composition of precipitation displayed a consistent trend of initial decrease followed by an increase. In the study area, the slope and intercept of the Local Meteoric Water Line (LMWL) formula were found to be lower compared to those of the Global Meteoric Water Line (GMWL).

(2) The amounts of straw returned to the field and irrigation affected the distribution of the soil water content. Under the same amount of straw returned to the field, with the reduction in irrigation, there was a gradual decline in the soil moisture content and the soil water content was lower than the wilting coefficient at the jointing stage. Under the same level of straw returning conditions, a decrease in the amount of irrigation resulted in a gradual decrease in the soil water content.

(3) At different growth periods and with the same irrigation amount, the straw returning amount had an effect on soil water $\delta^{18}O$ (because $\delta D$ and $\delta^{18}O$ were in good agreement with soil water isotopes). The $\delta^{18}O$ value of soil water exhibited a gradual decrease with increasing soil depth during various growth stages of wheat. The maximum $\delta^{18}O$ value of the soil water at each growth stage was distributed at 10 cm. Under identical irrigation conditions, an incremental rise in the $\delta^{18}O$ value was observed, as the straw returning amount increased from 0 to 20 cm and decreased gradually with the increase in the straw returning amount from 20 to 80 cm.

(4) By means of direct comparison, it was concluded that the soil water from the 10, 20, 30, 40, and 60 cm soil layers and their adjacent soil layers was preferred for use by winter wheat at the jointing stage under different treatments. In contrast, winter wheat at the flowering stage preferentially utilized soil water from the 20, 30, and 40 cm soil layers and their adjacent soil layers. The following results obtained via the direct comparison method and multivariate mixed model were consistent: during the jointing and flowering stages, the primary source of water for winter wheat was the soil water stored within the 0–30 cm soil layer; and the contribution of soil water below 30 cm to the water consumption of winter wheat from the jointing to the flowering stage tended to decrease.

**Author Contributions:** Conceptualization, L.W., X.Z., Y.C., K.Z., C.Z. and Q.L.; data curation, L.W. and Y.C.; formal analysis, L.W.; funding acquisition, L.W.; investigation, L.W., X.Z. and Y.C.; methodology, L.W.; project administration, L.W.; resources, Q.L.; software, K.Z.; supervision, Q.L.; validation, Q.L.; visualization, Q.L.; writing—original draft, L.W.; writing—review and editing, K.Z. and C.Z. All authors have read and agreed to the published version of the manuscript.

**Funding:** This study was financially supported by the National Natural Science Foundation of China (No. 52209049 and No. 51879066), the Natural Science Foundation of Hebei Province (No. E2019402468) and the Science and Technology Research and Development Program of Handan (No. 21422093247).

**Data Availability Statement:** The data presented in this study are available on request from the corresponding author.

**Conflicts of Interest:** The authors declare no conflict of interest.

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
