# Peer review of "Soil Moisture Contribution to Winter Wheat Water Consumption from Different Soil Layers under Straw Returning"

_agronomy, doi:10.3390/agronomy13112851_

Round 1
Reviewer 1 Report
Comments and Suggestions for Authors
Comments
General comment
For the whole manuscript use the same designation. For example: area 1 to 9, or S1V1 and so on, check table 3 and 4 and figure 9.
Specific comments
Materials and methods
1. L101: August (capital letter)
2. L101-105: you should decide either you abbreviate the months (Dec-Jan) or you write fully.
3. L125: since the study was already conducted, why do you affirm that soil samples will be collected?
4. L136: crops “were” collected. The epidermis,,, “was” removed
5. L142: automatic rain gauge “was”
6. L149: RSA and RST are the same as Rsa and Rst? If yes, why are you using different abbreviations? (one with capital and one with small letter?)
7. L153: how did you manage the statistical difference between the treatments?
Figures and tables
1. Figure 2: Are the dates in x axis correct? (for instance, in 2a, you have 01/12/2020 then 21/12/2020 and then 10/12/2020) for 2b there is also the same mistake.
2. Figure 3: What is the meaning of LMWL and GMWL. You should specify in the figure caption description although its already written in the above text. You should provide all the information to the readers in order to let give them much job to go through the text to understand you figures. (Same for figure 4)
3. Figure 5, 7, 8 and 9: Since you use same color for all dates in all areas, there is no need to repeat for each graph the legend. In one is enough.
4. Figure 6: to facilitate the comprehension, I suggest you to use the same color for the same treatment even in different stages.
Does the title of figure 6 reflects the information in the above graphs as well as the information in the 3.3 section?
.
5. Table 2: Is there any statistical difference between the treatments?
6. Table 3: Please use same model for the dates for the whole manuscript, (4-14 or April 14)
7. Table 4: Is there any statistical difference among the treatments?
Comments on the Quality of English LanguageSince the study was already conducted it is better to rewrite the manuscript in the past and not future.
Reviewer 2 Report
Comments and Suggestions for Authors
Please see attached pdf

The language is fine
Reviewer 3 Report
Comments and Suggestions for Authors
Dear Editor-in-Chief,
Thank you very much for the invitation to evaluate the manuscript (Contribution of soil moisture from different soil layers to water consumption of winter wheat in the dry land under straw returning). The work contains relevant and important information for the region. However, I believe that major changes must be made to the text. Some symbols are not appropriate, for example, the “~”, I believe it should be “-”. The Results section sometimes looks like a Discussion. There is a lack of numerical basis in the results. The Discussion section must contain works that support the study and explain its findings. Although the Conclusion is coherent, I recommend changes with information with greater impact, as it stands, it seems like a repetition of results.
Below are some observations to help improve the manuscript.
Line 19: Please change to: show that: (1) the
Line 35: Please, the Conclusion section should be clearer and with an innovation.
Line 39: Are these characteristics of air temperature?
Line 47: This information is very relevant, but is it “30% to 50% of irrigation water” or “30% - 50% of irrigation water”?
Line 61: What were these studies? Add at least three studies!
Line 107: Please add the meaning of DEM.
Line 115: How was the field capacity of the soil determined?
Line 144: Whose methodology is it?
Line 185: This date is confusing. Please add the year.
Line 187: The “~” sign means approximation, I think this symbolism is not appropriate.
Line 193: How much smaller was it?
Line 249-257: All these remarks are sounding like an Argument. The results must present findings and numerical values.
Line 260: The “()” symbol is being used improperly.
General comment from the Discussion section:
The authors should add more explanations and comparative studies to support their findings.
The Conclusion is supported by the results and objective of the study.
Comments on the Quality of English LanguageSeveral adjustments to English are necessary. The text has problems with the use of commas and incorrect words.
Round 2
Reviewer 2 Report
Comments and Suggestions for Authors
Please see attached.

Language is fine.
Author Response
Thank you very much for your valuable feedback on our manuscript, and we hope that our revisions will receive your approval. Thank you again for your valuable feedback and suggestions. The specific modifications are as follows:
Comment 1: L162: Explain what is direct comparison.
Response: Thank you very much for your review comments. Revised and highlighted in blue on line 165 of the text.
Comment 2: L162: write out the mixed models or explain it. tons of models can be called as mixed models, what is the specific mixed model that you are referring to?
Response: Thank you very much for your review comments. Revised and highlighted in blue on line 162 of the text.
Comment 3: L162 too vague, please specify and add more details.
Response: Thank you very much for your review comments. Revised and highlighted in blue on line 163 of the text.
Comment 4: L167: add the justification for these numbers.
Response: It was our negligence to cause confusion for you. Setting the source increment and mass balance tolerance parameters to 2% and 1% is to balance accuracy and flexibility in the IsoSource model to ensure that the calculation results of the model are in line with the actual situation.
Comment 5: L170: specify the models.
Response: Thank you very much for your review comments. Revised and highlighted in blue on line 172-173 of the text.
Comment 6: L190: from XXX to XXX.
Response: Thank you very much for your review comments. Revised and highlighted in blue on line 193 of the text.
Comment 7: L193: double check the legend to make sure it is correct.
Response: Thank you very much for your review comments. Revised and highlighted in blue on line 195 of the text.
Comment 8: L197: too absolute.
Response: Thank you very much for your review comments. Revised and highlighted in blue on line 199 of the text.
Comment 9: L214-216: Move to discussion.
Response: Thank you very much for your review comments. Revised and highlighted in blue on line 357 of the text.
Comment 10: L218: Add citation to support your opinion.
Response:Thank you very much for your review comments. Revised and highlighted in blue on line 363 of the text.
Comment 11: L235-236: Delete.
Response : Thank you very much for your review comments. Deleted based on comments.
Comment 12: L252-253: Add citation and move it to discussion.
Response :Thank you very much for your review comments. Revised and highlighted in blue on line 378-380 of the text.
Comment 13:L437: Do not repeat the result.
Response : Thank you very much for your review comments. Revised and highlighted in blue on line 435 of the text.
Comment 14: L438-440: This is more like result rather than conclusion, rephrase the sentence.
Response :Thank you very much for your review comments. Revised and highlighted in blue on line 435-438 of the text.
Reviewer 3 Report
Comments and Suggestions for Authors
Dear Editor-in-Chief,
Thank you for providing a revised version of the manuscript. The study is relevant, and the authors did a good job. There were great efforts to improve the text, but some corrections are still necessary.
Below are some observations:
Line 109: Please adjust the variable name. Air temperature is written next to it.
Line 214: Change to “follows: first, returning”
Line 238: The meaning of the letters is unclear.
Line 370: Camellia oleifera is a species, so its name should be in italics.
Line 421: The citation is wrong, please provide the citation number.
Comments on the Quality of English LanguageLittle adjustments.
Author Response
Comment 1:Line 109: Please adjust the variable name. Air temperature is written next to it.
Response: Thank you very much for your comments. It has been revised in line 110 of the text and highlighted in blue.
Comment 2:Line 214: Change to “follows: first, returning”
Response::Thank you very much for your comments. It has been revised in line 215 of the text and highlighted in blue.
Comment 3:Line 238: The meaning of the letters is unclear.
Response:Thank you very much for your comments. It was our negligence to cause confusion, and we have made additional modifications and highlighted them in blue in lines 240 to 243 of the text.
Comment 4:Line 370:Camellia oleiferais a species, so its name should be in italics.
Response:Thank you very much for your comments. It has been revised in line 375 of the text and highlighted in blue.
Comment 5:Line 421: The citation is wrong, please provide the citation number.
Response:Thank you very much for your comments. Reference [37] has been added to the article and highlighted in blue.
Comments on the Quality of English Language
Response:Thank you very much for your valuable feedback on our manuscript. We have conducted language editing of the entire text based on the website provided by the editor. Thank you very much for your enthusiastic cooperation. We hope that our modifications can receive your approval. Thank you again for your valuable feedback and suggestions.